# Anthropogenic Landforms Derived from LiDAR Data in the Woodlands near Kotlarnia (Koźle Basin, Poland)

**DOI:** 10.3390/s22218328

**Published:** 2022-10-30

**Authors:** Jan Maciej Waga, Bartłomiej Szypuła, Kazimierz Sendobry, Maria Fajer

**Affiliations:** 1Institute of Earth Sciences, Faculty of Natural Sciences, University of Silesia in Katowice, ul. Będzińska 60, 41-200 Sosnowiec, Poland; 2Independent Researcher, 43-100 Tychy, Poland

**Keywords:** relief, forest, cultural heritage, Koźle Basin, LiDAR, economic activity, digital databases

## Abstract

Unlike farmland or urban areas, forests have long been regarded as environments that favour the preservation of valuable geological and historical sites. However, due to invasive forestry methods, the implementation of large investment projects and the development of mining, they are increasingly no longer safe spaces for the relics of human activities recorded in landforms. Data collection, including using LiDAR technology, presents an opportunity to preserve knowledge about these landforms. Through the analysis of shaded images of a 37-hectare woodland area near Kotlarnia, landforms of various ages documenting 21 different human activities were identified, including remnants of reforestation activities, the expansion and modification of hydraulic structures and road infrastructure, charcoal burning and tar distilling, exploitation of mineral resources and military activities. The results of the remote sensing work were verified in the field.

## 1. Introduction

Until recently, forests were regarded as areas favourable to the preservation of historical sites, settlement relics, traces of human economic activity or war events. From the archaeological point of view, they were even treated as terra incognita [1]. The presence of traces of human activity in forest areas is clear from the analysis of landforms.

Unfortunately, due to increased use of invasive methods in these areas (mainly preparing soil for new planting, but also implementing large investment projects or digging large open-cast mines), woodlands have ceased to be safe places for the relics of human activity recorded in the relief of the land. The only chance to preserve knowledge about these landforms is to archive the relevant information, and today’s archives take the form of digital databases (e.g., [2,3,4]). Multiple opportunities to gather such information are provided by aerial remote sensing [5,6], methods involving the use of unmanned aerial vehicles (UAVs) [7,8,9] and, in particular, the analysis of high-resolution terrain models obtained from LiDAR laser scanning and subsequently verified by field methods (e.g., [10,11,12]). Although the Earth Sciences have used airborne laser scanning (ALS) in surveys of forest-covered areas for more than 20 years [13,14], it remains an attractive research method that is constantly being improved [15,16].

The purpose of this study was to identify traces left by human activity in the relief of woodlands in the area where sand is to be mined in the Kotlarnia open-cast mine in the Koźle Basin, and also to test the utility of hillshade relief models of different resolutions for this type of research. The distribution of man-made landforms and their morphological and morphometric features were analysed against the background of the natural relief and geological structure. Basic field documentation of those landforms was also drawn up and the relevant data were linked to existing information resources. 

## 2. Study Area

The study area, measuring 0.37 km^2^, is a wedge-shaped inselberg, extending along the north–south axis, separating the 2nd and 3rd sand mining fields in the Kotlarnia sand mine in the Bierawa municipality, Opole Province (Figure 1). According to the physico-geographical division of Poland, this area is located within the Koźle Basin, which is part of the Silesian Lowland [17]. Geomorphologists consider it to be either part of the Silesian Lowland or the Subcarpathian Basins [18]. 

In the Koźle Basin, there are morphological levels associated with the deglaciation of the Odra ice sheet and several river terraces. The study area includes a denuded slope of the Bierawka River valley, which consists of glacial and fluvioglacial formations from the Odra Stage, and a river terrace from the Vistula Stage, which consists of sands and silts [19,20]. In the uppermost layer of these sediments, there are aeolian cover sands, whose thickness ranges from several dozen centimetres to 2 m, and numerous dunes [21]. Prior to sand mining, there were surface depressions filled with organic sediments supported by sand dunes, former braided river channels, denudation valleys with small gradients and deflation basins in the area. Today, only fragments of those remain in the part where mining has not started yet (Figure 2).

On the sandy bottom of the Koźle Basin, podzolic and rusty soils have developed, which are covered mainly by pine forests with some oaks present. In the past, numerous shallow depressions in the area contained water (both in the form of water bodies and wetlands). Currently, due to the deliberate draining of forest areas using drainage ditches, and aggregate mining activities, surface waters and shallow ground waters have completely disappeared in both Kotlarnia and Dziergowice [22]. These waters are now discharged directly into the Bierawka River or into mine workings.

## 3. Materials and Methods

### 3.1. Materials

In the first stage (in-house studies), the authors analysed the hillshade relief raster of the selected area on the polska.e-mapa.net website [23], generated from a DEM with a resolution of 1 × 1 m. Digital elevation models were subsequently prepared with resolutions of 0.1 × 0.1 m and 0.05 × 0.05 m. LAS point clouds (compliant with the standard in [24]) derived from aerial laser scanning (ALS) with a minimum density of 12 points/m^2^ and an average vertical accuracy of <0.1 m were used. ALS data were acquired with a Leica ALS70 scanner during a flight on 9 April 2019. All data were in the EPSG:2180 coordinate system. Class 2 (ground) points from the first and second reflections were selected from *.las files. On the basis of these, using the Global Mapper [25] software with default settings (create an elevation grid for use in analysis—binning, minimum value—DTM), a digital elevation model was created, and subsequently a hillshade relief raster of the land surface was constructed with the same resolution as the source model (i.e., 0.1 × 0.1 m and 0.05 × 0.05 m), with standard illumination settings (azimuth 315°, altitude 45°). An orthophotomap of the study area from the polska.e-mapa.net website was also analysed.

In order to determine the environmental conditions and examine the historical background of the study area, 1:50,000 maps were analysed, which included geological, environmental protection and hydrographic maps, and a soil map from the Opolskie w Internecie geoportal [26], together with information from the Central Geological Database of the Polish Geological Institute [27]. Forest geomatics information, including tree stand age and therefore the timing of the activities that affected landforms in the area, was obtained from the Forest Data Bank portal [2]. Available historical sources were also used, including old maps and oral accounts by local residents.

### 3.2. Methods

The landforms selected for further study were located in the field using a GPSMAP (Olathe, USA) 66S receiver. Morphometric work in the field was conducted using an RTK Leica Viva CS10 high-precision GPS with an average measurement accuracy of 1 cm (horizontal) and 1.3 cm (vertical) and using the Nikon Forestry Pro II (Nikon Corporation, Tokyo, Japan) and Leica DISTO X310 (Leica Geosystems, Heerbrugg, Switzerland) laser rangefinders, a surveying staff and tape measure.

Geological conditions were identified on the slopes of natural formations and on the walls of the Kotlarnia and Dziergowice mine workings. Near-surface soil conditions were tested with a 1-metre sampling stick. Dendrochronological measurements were conducted on felled tree trunks or on cores collected with a Pressler auger. 

The species of trees burnt in a pile of charcoal were identified by analysing 100 tree crumbs 7–40 mm long [28]. They were subject to taxonomic identification on the basis of the preserved wood anatomy. The samples were examined in three planes—transverse, tangent and radial—in the range of magnifications from 100× to 500× in reflected light using the Olympus BX53M metallographic microscope. The Stream Essentials 2.1 microimaging software was used. The observed anatomical features were then compared with contemporary comparative specimens and data from the patterns contained in the tree species recognition keys. 

Methods used in work on similar topics related to woodlands have been comprehensively presented, inter alia, in articles by collective research teams [10,29,30].

## 4. Study Results

Principally using LiDAR data, an overview of various elements of terrain relief was obtained, indicating signs of human activity in the area where sand is to be mined by the Kotlarnia mine.

### 4.1. Landforms Documenting Forest Management Activities

The surface morphology of the study area shows the effects of the different reforestation methods implemented after thinning and clearcuts, and of the treatments carried out during tree growth [31,32]. In addition to natural reforestation, plantings were conducted, either without or with soil preparation. Where the no-till method was used and, at most, plots for groups of trees or spots for individual trees were created, the ground surface is closest to the original relief (Figure 2a). In the study area, those treatments were applied within control units (subdivisions of forest divisions) established 95–100 years ago [2]. Clearcuts from 30–55 years ago were ploughed with a forest plough, although this is also locally evident in parts of units that were planted 80 years ago. The originally shallow (around a dozen centimetres) furrows dating back to older reforestations from 80 years ago are ca. 1.1 m wide, while the later ones, from 55 years ago, are 1.5 m wide (Figure 2b).

In the northern part of the area, strips of land, which are more than 30 years old and were created by deep, full tillage conducted at two depth levels, are visible—with the narrower ones being 2.5 m wide (Figure 2c), and the wider ones 5 m wide (Figure 2d). The furrows separating these wide strips are deeper, originally reaching at least 0.6 m. They may have been created with a special large double mouldboard plough (the so-called Matuszczyk model) [33] and could have exceeded 0.8 m in depth. This area included coniferous forest sites with groundwater levels artificially lowered by the sand mine. After a fire in the 1970s, the area was overgrown with reed grasses. Locally underlain by hardpan and with soils contaminated by industrial emissions from Kędzierzyn, it must have been difficult to reforest. Today, the iron oxides present in the hardpan still lend a rusty colour to the formations visible in the wall of the sand mine workings (Figure 3).

An even, uncultivated plot in the middle of the forest is visible in the central part of the study area (Figure 2e), which used to be encircled by a tall fence. It was used to cultivate forage crops to feed forest animals in winter. It is currently occupied by about 30-year-old forest, which makes it difficult to detect its presence from the adjacent forest track. Technically, natural and artificial clearings within forests can be divided into temporary and permanent ones, but as is apparent from this example, the latter may also vanish from the ecosystem after several decades.

On the opposite side of the aforementioned forest track is a complex where remains of 11 dugouts are present (Figure 2f and Figure 4). One of those could have been used to store planting material, but due to the concentration and arrangement of the remaining dugouts, their use as a site for soldiers during World War II cannot be ruled out. This was confirmed by forestry supervision workers.

### 4.2. Forms Documenting Changes in Hydraulic Structures and Road Infrastructure

From the southeast to the northwest, the study area is crossed by the valley and channel of the stream that once used to drain water from the area situated between the Ruda and Bierawka Rivers to the Bierawka River near Korzonek. The stream must have exhibited significant flows in the past, at least periodically. During floods, its water overflowed onto the floodplain terrace, which contains traces of multichannel flows, the erosive formations created by floods and local marshes. Depressions between the dunes were waterlogged, which is evidenced by the presence of drainage ditches of various sizes (Figure 2g), a relic of a spring niche or a drained bog-spring (Figure 2h) and a World War II bomb explosion crater with a central peak characteristic of waterlogged areas (Figure 2i). The stream channel was eventually dredged and straightened. Today, it is dry and cut off on both sides by the tall slopes of the open-cast sand mine (Figure 2j). Surface waters flowing in the area between the Bierawka and Ruda Rivers were drained by two ditches (5 m and 9 m deep) running in the east–west direction south of the study area (Figure 2k). Currently, the system that drains surface waters from the study area only functions locally and to a very limited extent. Water is only drained via this system during heavy rainfall, usually in areas underlain by layers of silts (Figure 5), briefly forming hanging groundwater tables there. The main water table in the area has been lowered by as much as 15–17 m as a result of sand mining.

As far back as 300–400 years ago, hydraulic structures were erected in the area and bridges were constructed [22], whose approach embankments were often reinforced with fascine and metallurgical slag sourced from nearby forges [34]. In the study area, a road junction was present close to one of these forges (Figure 2l), which was marked with a large erratic boulder for orientation (Figure 6). At the time when forest administration was reformed following the edict issued by Frederick II in 1756, forests in the state that he ruled were divided into rectangular divisions, forest tracks were marked out and new roads were constructed along these tracks [22,35,36]. Sometimes, the divisions were adjusted to fit the former boundaries of forest properties and important transport routes. This was also the case with the road running through the study area from Lubieszów and Dziergowice to Ortowice, Kotlarnia and Tworóg Mały. Some sections of old roads fell into disuse and were overgrown with forest. However, their traces were recorded as incisions in dune slopes (Figure 2m) and embankments in terrain depressions (Figure 2n). These are clearly discernible in shaded terrain models. Today, due to the fact that mineral extraction takes place in the study area, the importance and use of the various roads has further changed. New access and haul roads have been constructed (Figure 2).

### 4.3. Remnants of Charcoal Burning and Tar Distilling

As demonstrated by the studies of remnants of charcoal burning and tar distilling carried out using DEM, traces of charcoal piles are common finds in Polish forests [37]. This is confirmed by historical, cultural and environmental studies [22,36,38,39,40,41,42]. Charcoal pile remnants are particularly abundant in the vicinity of former metallurgical industry districts, although there were many other customers who purchased the charcoal and tar produced. In the basins of the Mała Panew, Stobrawa, Kłodnica and Czarna Rivers, where numerous iron forges operated, more than 200,000 such piles have been identified [43]. Forests growing on the banks of the Ruda and Bierawka Rivers played a similar role as a raw material base for manufacturing and even for the energy and chemical industries. The studies conducted in Germany demonstrated morphological variation between charcoal pile bases (e.g., [44]). Charcoal piles surrounded by ditches have been described in Greater Poland [45], and similar structures have been found in Brandenburg [46]. In Upper Silesia and Lesser Poland, charcoal piles with several depressions along their circumference are particularly numerous [26,43]. Those depressions were created by the removal of earth in order to cover the log pile, and later served to collect water and other liquid products leaching from the wood during distillation. In the bottom sections of the charcoal piles in Wymysłów near the Katowice district of Panewniki, Mr Józef Mysłowski pointed out uncovered beams that probably formed a kind of a grate, or a drainage system facilitating the collection of these liquids from the charcoal pile. 

Apart from charcoal, which was commonly used as a fuel, filtering agent and medicine (charcoal burning), the thermal processing of wood yielded the widely used tar, pitch, rosin and turpentine (tar and pitch distilling), and ash, which was rich in elements and chemical compounds (ash production) and was used in the production of glass, soap, medicines and fertilisers. Potash and calcium were particularly important components of ash. Appropriate tree species and wood distillation technologies have been used for a very long time to obtain specific part-processed raw materials and products [47]. This can be approximately determined from the remains of charcoal piles, which either were or were not equipped with a drainage and collection system for the liquid fraction, and by precisely analysing the species of charcoal [48] and conducting geochemical studies of the substrate, which contains organic matter [44,49,50,51,52].

In the study area, relics of 61 charcoal piles with diameters ranging from 7 to 13 metres were found. Most of these were located in its northern part. Some are well preserved, which indicates their relatively young age [36]. The bases of three such charcoal piles, surrounded by several pits, are located in the middle of the northern part of the study area (Figure 2o). Two of the landforms also contain relics of surrounding ditches, and the third was partially damaged by a bomb explosion during World War II (Figure 7). The analysis of charcoal collected from this pile demonstrated that only pine logs were charred there. This corresponds with the presence of pits along the circumference of the base of each charcoal pile, which were dug in order to collect the liquid fraction from the distillation process.

### 4.4. Landforms Left by Mining Mineral Resources

In the study area, there are traces of sand mining for both local needs (related to the construction of the road system) and external needs (related to coal mining and construction). Gravel was also extracted. The first example concerns the use of sand from the road that cut through a dune in order to construct the upper part of the embankment crossing the bottom of the aforementioned valley of the stream flowing from the area situated between the Bierawka and Ruda Rivers. The second example involves some of the most dramatic landscape transformations in Poland as a result of the construction of the open-cast sand mine, with the excavated sand formerly used as coal mine backfill. Currently, as hydroplastic backfill is no longer used to fill the voids left by deep coal mining, the mine extracts sand primarily for construction purposes. In the west, two working wall steps are visible (Figure 2p) alongside the cleared area prepared for mining (Figure 2r). A layer of humus has been removed from this area and deposited in the mine workings nearby. To the east, several years ago, humus was pushed onto oblong piles up to 5 metres high (Figure 2s and Figure 8), which formed a barrier limiting access to the edge of the high working wall. Traces of gravel mining have been found within the two moraine hummocks located north of the dune range and the stream valley (Figure 2t). This gravel could have been used in construction and road engineering.

### 4.5. Landforms Left by Military Activities

In 1944, the 15th USAAF conducted 18 mass bombing raids in the Kędzierzyn area, attacking the largest synthetic liquid fuel production centre in the Third Reich [53,54]. As a result of the effective smoke screens laid over the plants, high drop heights and intense anti-aircraft fire, many of the 39,137 bombs dropped failed to hit their targets, exploding in other areas including forests. The craters preserved there, created by 500-pound demolition bombs, range from 10 to 15 metres in diameter and are often more than 2 m deep; 250-pound bombs resulted in 7–10 m craters, and there are also smaller craters from unexploded ordnance [55,56,57]. Before the sand mine started to operate in Kotlarnia, the area had been soft and waterlogged due to the fact that the substrate consisted of marginal silts, which were impermeable to water (Figure 5). This is confirmed by the presence of a crater with a central peak (Figure 2i). The presence of such landforms in the strongly waterlogged areas of central Poland was described as early as the beginning of World War II by Trusheim [58]. Seven craters created by unexploded aerial bombs were found in its vicinity, with some of the ordnance removed and neutralised by sappers. The high rate of unexploded ordnance in this area was also due to fact that detonator parameters were adjusted so that the bombs would explode upon hitting concrete and steel industrial and transport facilities. The arrangement of craters suggests that they were created during a raid on the IG Farbenindustrie plant in Heydebreck. 

The study area also contains a rare example of a well preserved dry crater, most probably formed by the explosion of two bombs that hit the same spot. This is evidenced both by the unusual size of the landform (up to 15.5 m in diameter and 3 m in depth) and by the shape of its slopes, which include a convex recess—a sort of an inner half-ring within the slope (Figure 2u and Figure 9). 

This slump may have resulted from the edge of the older crater sliding after the second explosion. In waterlogged areas, this takes the form of arcs and rings on the bottoms of flooded craters [59]. 

At the top of one of the hummocks north of the dune range and the stream valley in the forest, there is a rectangular pit measuring 8 × 5 m, which is 0.5–0.7 m deep. It may have been originally formed in the location where gravel was dug; however, its regular shape suggests that it may have eventually served as a German observation and firing post during World War II. The surrounding forests were heavily exploited at the time [60], with considerable amount of logging conducted and greenwood planted. Anti-aircraft batteries and the posts that protected them and military field quarters were located among those greenwood patches.

The analysis of the shape of three bomb craters located near the aforementioned hummock with the observation and firing post indicates their partial remodelling (Figure 2w). The first crater, which is located on the northern edge of the mid-forest plot, has straight northern and eastern edges. The other, lying by the stream channel, has three straight slopes. In the third crater, located east of the hummock with the pit, the modelling of the south-western slope was initiated. The first two landforms, which are situated at lower elevations, could have been adapted as water reservoirs, and the third, for instance, as a storage depot. Similar modifications of bomb craters for defence and storage purposes were observed at Kędzierzyn in the vicinity of the former IG Farbenindustrie plant [59].

## 5. Discussion

The use of LiDAR data in geomorphological studies has undoubtedly been one of the most important technological breakthroughs of our times [61,62,63]. Shaded relief rasters of various resolutions obtained from laser scanning provide very good materials for studying forest land surface morphology, although—like other remote techniques—they require the operator to be familiar with the subject matter in question and with the operating methods [10,64,65,66]. It is also possible to use algorithms for the semi-automatic and automatic recognition of landforms with repetitive shapes—a so-called machine search (e.g., [67,68]). Shaded terrain rasters are generally available for a certain accuracy range [23,69], which allows the search for valuable sites within the country to be crowdsourced [10]. In locations with a high canopy density, ground scanners or surveying instruments need to be used in order to study forest areas, but such cases are relatively rare. These activities can be performed as part of field verification work and detailed studies.

Studies using LiDAR data make it possible to find previously unidentified objects or reinterpret features of objects that have been incorrectly interpreted to date [70,71]. 

In this era of economic expansion into forested areas, the collection of data using remote sensing methods is particularly important, as it enables us to learn the history of areas subjected to intensive cultivation and destined to be transformed as a result of open-cast mining of mineral resources or the construction of large projects. Owing to these studies, knowledge about such areas will not be irretrievably lost and can serve further research purposes.

An example of this type of work is presented in this article: in the study area, the rate at which the single, 1.4 km long working wall of the Kotlarnia sand mine progresses is around 25 m/year. The plant, together with the Górażdże Kruszywa company, has applied for amendments to the zoning plan for the Bierawa municipality and for permission to excavate sand from further fields with an area of approximately 1.5 km^2^ [72]. Together with the area already allocated to sand mining, this represents around 2.5 km^2^ of forest land that will cease to exist in a dozen years at the latest, together with many geoenvironmental elements and cultural remnants.

## 6. Conclusions

High-resolution terrain models, which are very important in engineering studies in forestry, can be effectively used thanks to LiDAR technology. The research conducted confirmed that the use of modern measurement techniques (ALS) and then the collection of high-resolution materials (DEM) is an invaluable source of basic data for the analysis of the local relief in forested areas. In addition to the purely documentary function related to the nature, spatial distribution and morphometry of individual landforms, it also enables us to interpret the course of events based on the spatial relationships between these landforms. Obviously, in situ studies are required to verify the information and conduct precise microrelief measurements.

New geoinformation tools, which have been successfully deployed by geomatics experts in state forests in the last two decades, the databases compiled by these experts and their cooperation with other research centres provide opportunities for the accurate documentation of heritage, including forest management heritage, and contemporary changes in the relief. This allows for better management of the various types of resources present in forested areas and their study. Nowadays, political and economic decisions concerning forestry may—and should—take into account both economic considerations and issues related to protecting and documenting the heritage recorded in the relief of forest areas. Woodlands are particularly important in this context, since such a wealth of natural and man-made forms, which can be revealed by laser scanning, is not found in agricultural or urban areas.

## Figures and Tables

**Figure 1 sensors-22-08328-f001:**
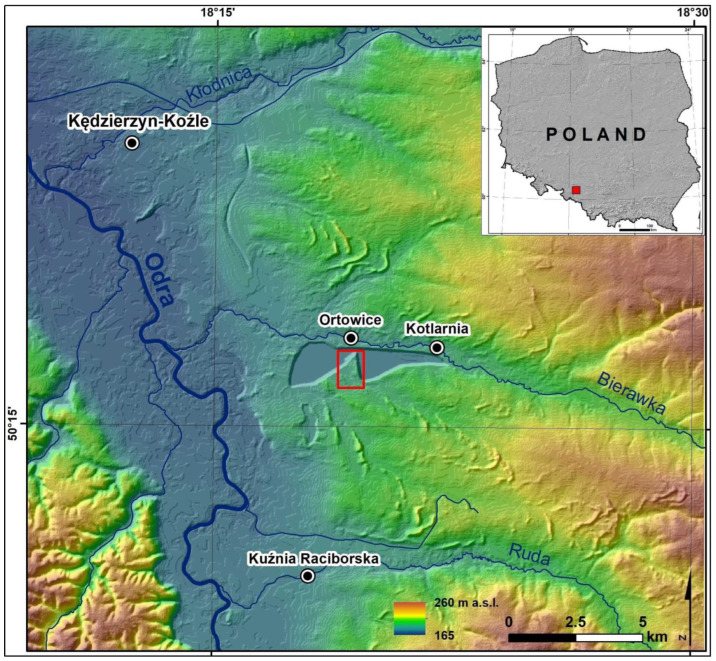
Location of the study area.

**Figure 2 sensors-22-08328-f002:**
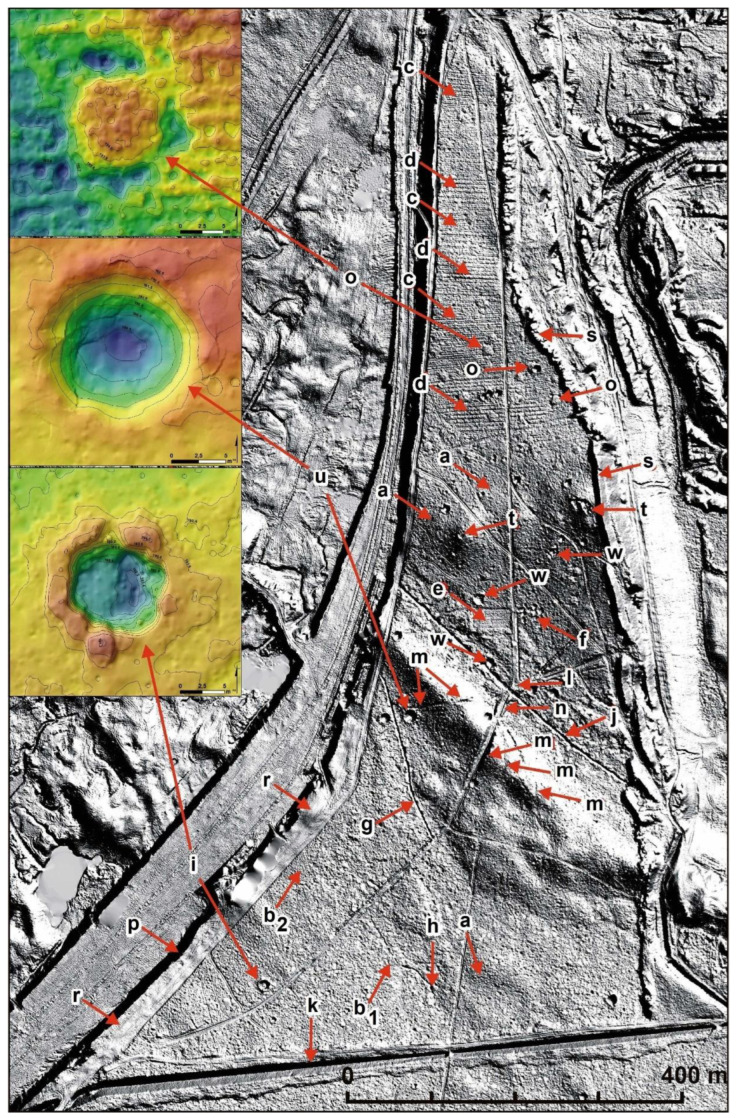
Study area: (**a**) forest area cultivated using the traditional no-till method around 100 years ago; (**b**) traces of shallow forest tillage from (**b1**) 80 and (**b2**) 55 years ago; (**c**) traces of deep tillage from 30 years ago; (**d**) traces of deep tillage from 30 years ago—deeper furrows breaking the hardpan; (**e**) mid-forest plot; (**f**) dugout complex; (**g**) drainage ditch; (**h**) spring niche or bog-spring; (**i**) bomb crater with a central peak; (**j**) dry stream bed; (**k**) deep ditch surrounding the sand pit; (**l**) forest road junction; (**m**) incision with a forest road; (**n**) road embankment; (**o**) charcoal pile remnants; (**p**) sand mine working wall; (**r**) strip with humus removed, prepared for sand excavation; (**s**) pile of humus overburden; (**t**) gravel extraction sites; (**u**) crater left by the concentric explosion of two bombs; (**w**) crater remodelled by human activity.

**Figure 3 sensors-22-08328-f003:**
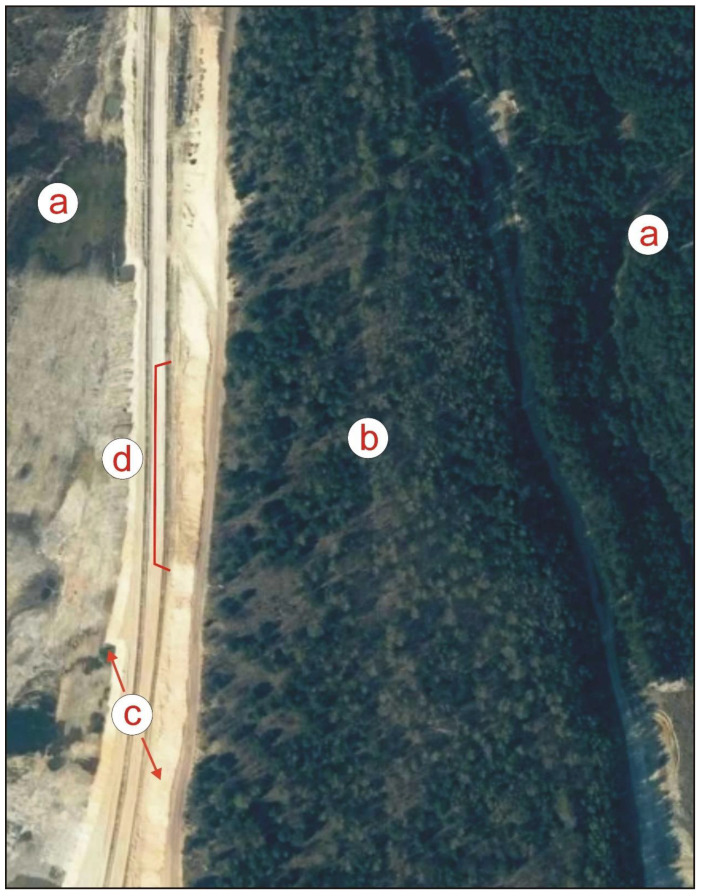
Northern part of the study area (source: e-mapa.net): (**a**) sand mine pit; (**b**) inselberg mined for sand—a complex comprising a Vistulian terrace and postglacial formations of the Odra Glaciation; (**c**) two working wall steps; (**d**) crumbling hardpan zone.

**Figure 4 sensors-22-08328-f004:**
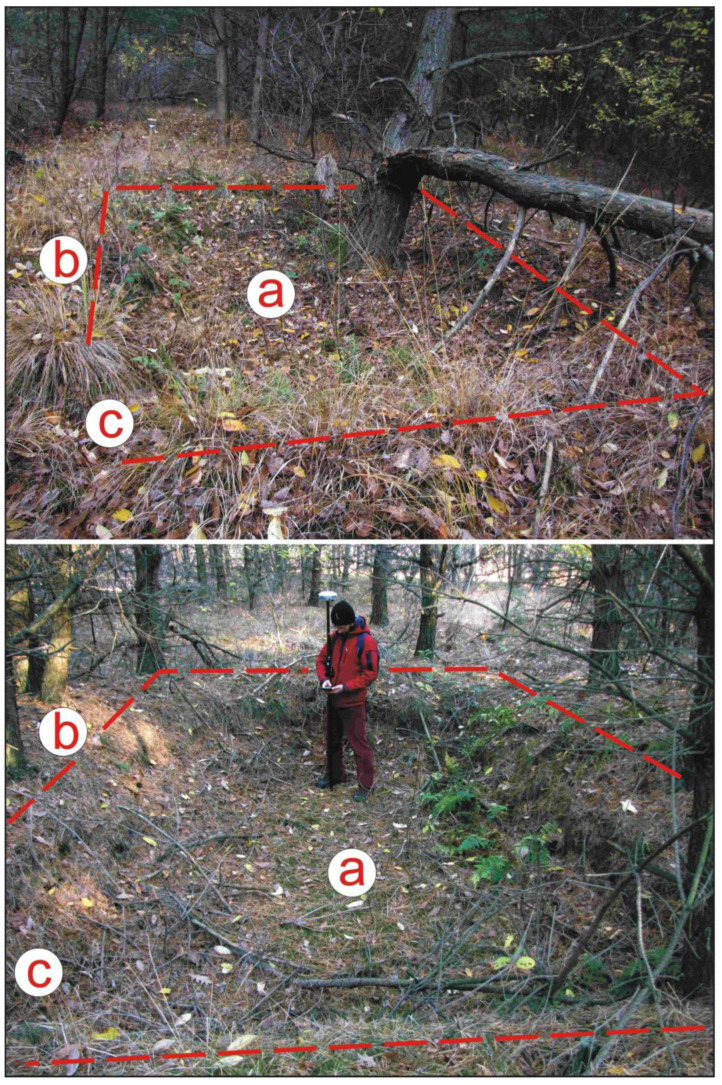
Two of the 11 forest dugouts: (**a**) depression; (**b**) surrounding embankment; (**c**) entry (photo: J.M. Waga).

**Figure 5 sensors-22-08328-f005:**
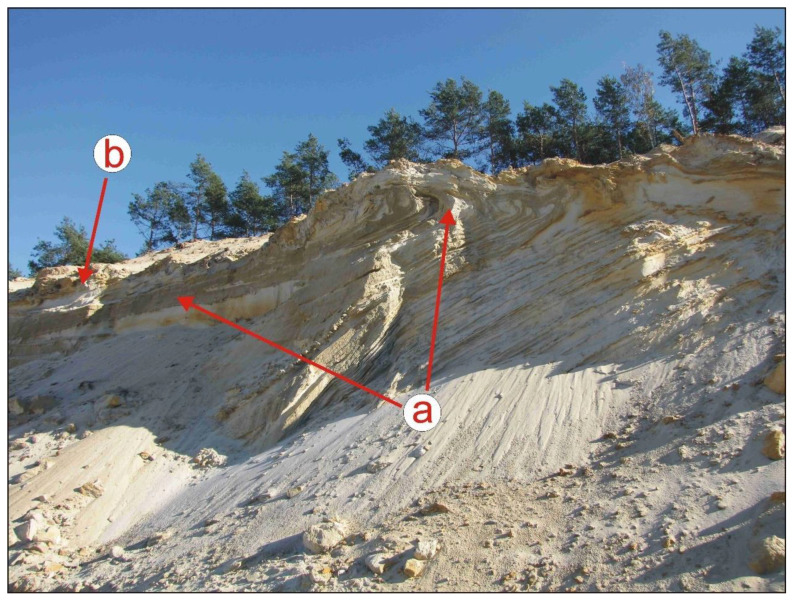
Layers of marginal sandy silts disturbed by the ice sheet: (**a**) in the open cast pit wall of the sand mine in the southern part of the study area; (**b**) a zone of former hanging groundwater level I (photo: J.M. Waga).

**Figure 6 sensors-22-08328-f006:**
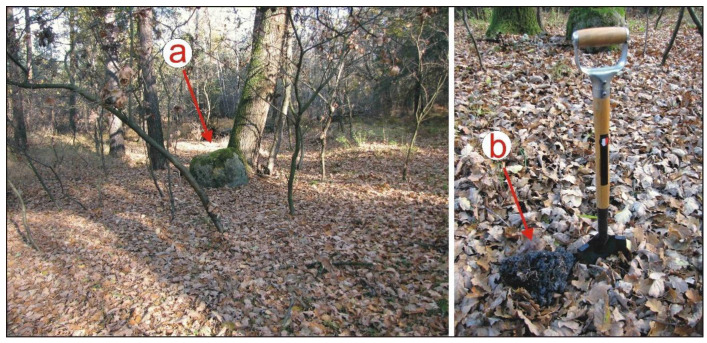
A junction of former forest roads marked by: (**a**) a large erratic boulder; (**b**) a crumb of smelter slag used to harden the road surface—central part of the study area (photo: J.M. Waga).

**Figure 7 sensors-22-08328-f007:**
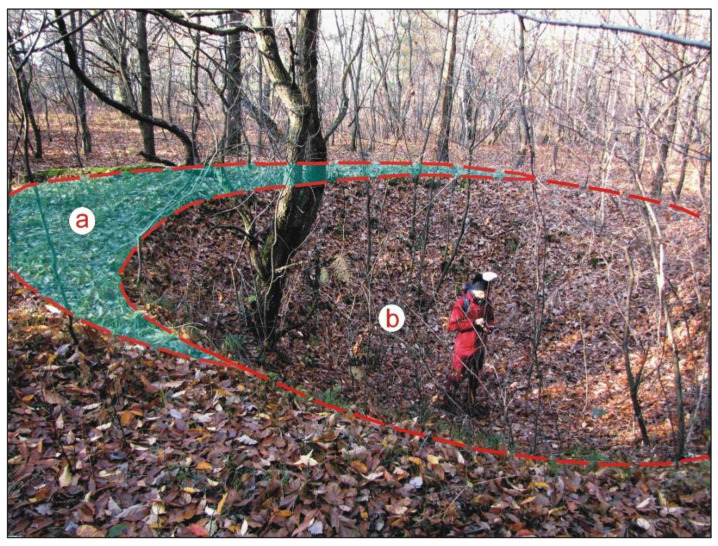
Remnants of a charcoal pile partially destroyed by a bomb explosion during World War II: (**a**) charcoal pile remnants; (**b**) bomb crater (photo: J.M. Waga).

**Figure 8 sensors-22-08328-f008:**
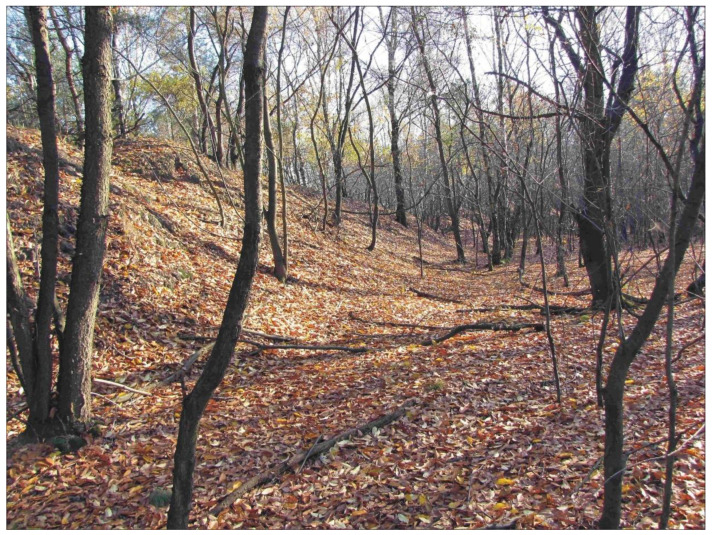
A pile of humus overburden removed from a sand mine field (photo: J.M. Waga).

**Figure 9 sensors-22-08328-f009:**
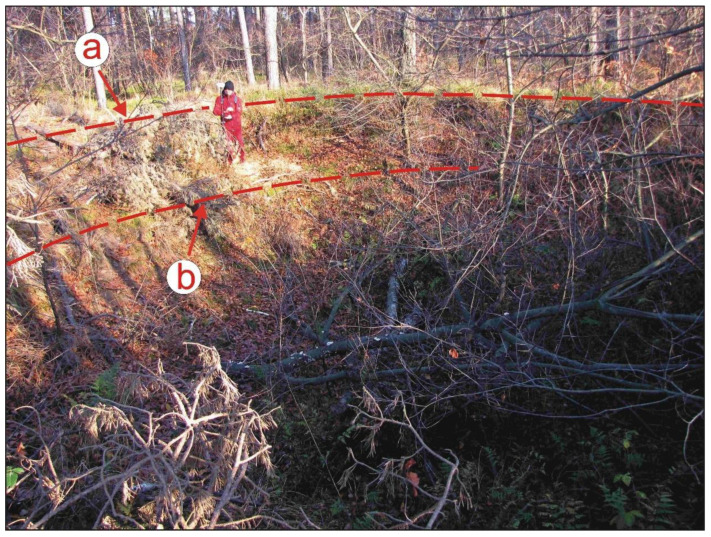
A large crater created by concentric explosions of two bombs during World War II: (**a**) crater edge; (**b**) edge of the slump formed after the second explosion (photo: J.M. Waga).

## Data Availability

Not applicable.

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
