# Peer review of "Anthropogenic Landforms Derived from LiDAR Data in the Woodlands near Kotlarnia (Koźle Basin, Poland)"

_sensors, 2022, doi:10.3390/s22218328_

Round 1

Reviewer 1 Report

This is a very interesting manuscript on using LiDAR data in geomorphological studies. The manuscript is clearly written and motivated and I believe will be of great interest to the readers of "Sensors". I only have some minor comments.

1) line 26: "[...] due to increased use of invasive methods in these areas"

2) line 35: Define the acronym. 

3) Figure 2: I think will be easier to read this figure if the labels will be in white (for contrast). 

4) Line 83: Remove "of the paper". 

5) There is no need for an Acknowledgements section?

Author Response

Dear Editor and Reviewers!

Firstly, we would like to thank you for all Your comments and suggestions. Owing them – our paper is better. We made all necessary changes, all of which are highlighted in yellow in the text.

Regarding Review Report No. 1:

Line 26: "[...] due to increased use of invasive methods in these areas"

This sentence has been corrected.

Line 35: Define the acronym. 

Acronym has been explained.

Figure 2: I think will be easier to read this figure if the labels will be in white (for contrast). 

This Figure 2 has been corrected – all labels has been changed.

Line 83: Remove "of the paper". 

We removed this fragment from the text.

There is no need for an Acknowledgements section?

Thank You, we completed Acknowledgements section: „The authors would like to thank Prof. Andrzej Czylok for the discussion and valuable comments on forest ecology and forestry, as well as the Director of the Institute of Earth Sciences at the University of Silesia in Katowice, Prof. Ewa Łupikasza, for financial support of the research. We would like also to thank two anonymous reviewers for their valuable comments, which made the article better.

Reviewer 2 Report

Dear authors,

Thank you for your perspective on the implementation of LiDAR technology in forest lands. My personal views on your work are as follows. I wish you success in your work.

(Line 21). Your introduction section needs to be expanded. Information about previous work can be added. The emphasis on the widespread/common use of LiDAR has been supported by the literature. However, studies similar to yours and their results can be mentioned. Adding some descriptive information will increase reader interest, as well as popularity of your paper.

(Line 46). The work area is explained in detail. Geographical locations are indicated. This section is enough.

(Line 82). You may need to separate the Material and Method sections into subheadings. The separation of equipment and method would make more sense for readers. For example, the Workspace is also your material. Separate the equipment and data you used in the study. You will be advised to edit this section. Line 120-121. "1:50,000 maps were analyzed" can be written as 1:50000 scaled maps given in the sentence. It seems that you have determined the change points through observation. If it was carried out by a study involving automatic detection (mathematical model), you can specify it.

The findings you have obtained in the determination and analysis of the spatial points that have been altered by human. Descriptive figures are sufficient.

(Line 380-382). As a suggestion. In the conclusion, your first sentence can be modified. For example, "High resolution terrain models, which are very important in engineering studies in forestry, can be used effectively thanks to LiDAR technology".

I hope my suggestions will contribute to the improvement of your work.

Regards,

Author Response

Dear Editor and Reviewers!

Firstly, we would like to thank you for all Your comments and suggestions. Owing them – our paper is better. We made all necessary changes, all of which are highlighted in yellow in the text.

Regarding Review Report No. 2:

Line 21. Your introduction section needs to be expanded. Information about previous work can be added.

Thank You, we have expanded the Introduction section and added the necessary references.

Line 46. The work area is explained in detail. Geographical locations are indicated. This section is enough.

Thank You.

Line 82. You may need to separate the Material and Method sections into subheadings.

Thank You for the suggestion. We decided to divide Chapter 3 into two subchapters: 3.1 Materials, 3.2 Methods. We agree, now it looks better.

Line 120-121. "1:50,000 maps were analyzed" can be written as 1:50000 scaled maps given in the sentence. It seems that you have determined the change points through observation. If it was carried out by a study involving automatic detection (mathematical model), you can specify it.

No, there was no mathematical modeling here. The 1:50,000 map scale is the scale commonly used in Poland for detailed thematic studies (geological, hydrographic, sozological, etc.).

Line 380-382. As a suggestion. In the conclusion, your first sentence can be modified. For example, "High resolution terrain models, which are very important in engineering studies in forestry, can be used effectively thanks to LiDAR technology".

Thank You for this suggestion, we have added this sentence to the Conclusion section.
